

# Estimating Annual Water Storage Variations Using Microwave-based Soil Moisture Retrievals

Wade T. Crow[1], Eunjin Han[2], Dongryeol Ryu[3], Christopher R. Hain[4], Martha C. Anderson[1]

[1] USDA Hydrology and Remote Sensing Laboratory, Beltsville MD, USA
[2] International Research Institute for Climate and Society, Columbia University, NY, USA
[3] University of Melbourne, Melbourne, Victoria, Australia
[4] Earth System Science Interdisciplinary Center, University of Maryland, College Park MD, USA

*Correspondence to*: Wade T. Crow (Wade.Crow@ars.usda.gov)

**Abstract.** Due to their shallow vertical support, remotely-sensed surface soil moisture retrievals are commonly regarded as being of limited value for water budget applications requiring the characterization of temporal variations in total terrestrial water storage ($S$). However, advances in our ability to estimate evapotranspiration remotely now allow for the direct evaluation of approaches for quantifying annual variations in $S$ via water budget closure considerations. By applying an annual water budget analysis within a series of medium-scale (2,000-10,000 km$^2$) basins within the United States, we demonstrate that, despite their clear theoretical limitations, surface soil moisture retrievals derived from passive microwave remote sensing contain significant information concerning relative inter-annual variations in $S$. This suggests the possibility of using (relatively) higher-resolution microwave remote sensing to enhance the spatial resolution of $S$ estimates acquired from gravity remote sensing. However, challenging calibration issues regarding the relationship between $S$ and surface soil moisture must be resolved before the approach can be used for absolute water budget closure.

## 1 Introduction

Within the past decade, the analysis of data products from the Gravity Recovery and Climate Experiment (GRACE) satellite mission (Tarpley et al., 2004a; 2004b) has led to an enhanced appreciation of the role played by inter-annual variations of total terrestrial water storage ($S$) within the terrestrial water budget (Chen et al., 2009; Rodell et al., 2007; Syed et al. 2008). However, the application of GRACE $S$ retrievals is potentially limited by their extremely coarse spatial resolution (~200,000 km$^2$). In contrast, microwave-based surface soil moisture ($\theta$) retrievals can be obtained at relatively finer resolutions (typically ~1,000 km$^2$). However, such



retrievals are hampered by both shallow vertical support (reflecting soil moisture conditions only in the top several centimetres of the soil column) and substantially-reduced accuracy for dense vegetative cover. As a result, they are generally assumed to be of limited value for examination of $S$ variations and commonly neglected in water budget studies. However, recent empirical work demonstrates that microwave-based $\theta$ retrievals are well

correlated with GRACE-based $S$ estimates in certain regions (Abelen et al., 2013; 2015). This suggests that $\theta$ retrievals retain some value for water-balance studies - particularly at spatial scales finer than the resolution of GRACE products.

Confirming such potential will require the availability of accurate terrestrial water flux variables. Recent progress in the remote sensing of $S$ and $\theta$ has been mirrored by the increased consideration of satellite-derived

evapotranspiration ($E_T$) retrievals in a water balance context (Senay et al., 2011; Hain et al., 2015; Hendrickx et al., 2016; Wang-Erlandsson et al., 2016). In particular, when combined with precipitation ($P$) and basin-outlet steam flow ($Q$) measurements, satellite-derived $E_T$ estimates can be used to verify estimates of $S$ variations ($dS/dt$) obtained from various independent sources (Han et al., 2015). This opens up the possibility for the objective "top-down" evaluation of $dS/dt$ estimates obtained from various remote sensing sources and the opportunity to

empirically confront "bottom-up" expectations for these products based solely on theoretical considerations.

Here, we combine $E_T$ estimates acquired from thermal infrared (TIR) remote sensing with ground-based $Q$ and $P$ measurements to evaluate the water balance performance of passive microwave (PM) estimates of annual $dS/dt$ for a set of medium-scale (2,000-10,000 km$^2$) river basins within the United States. The analysis will focus on two primary tasks: 1) evaluating the suitability of existing $E_T$, $Q$ and $P$ data products to accurately estimate $dS/dt$

and 2) empirically investigate the ability of inter-annual $dS/dt$ estimates (acquired from microwave remote sensing of soil moisture) to close the inter-annual terrestrial water balance. As discussed above, this particular application of $\theta$ is arguably inconsistent with their known theoretical limitations. Therefore, our focus will be on empirically measuring their ability to provide $dS/dt$ closure within an annual water budget analysis and examining how these empirical results fit with *a priori* theoretical expectations.

Section 2 describes the water balance data sets and study basins. Section 3 examines the ability of existing flux and storage products to close the terrestrial water balance closure within a set of larger-scale (150,000-1,000,000 km$^2$) hydrologic basins where GRACE-based $dS/dt$ can be directly utilized (see task #1 defined above). Based on verification results in Section 3, Section 4 derives a technique for estimating $dS/dt$ from microwave remote



sensing and evaluates the ability of microwave-based *dS/dt* to close the terrestrial water balance within a second set of medium-scale (2,000-10,000 km$^2$) basins (see task #2 defined above). Results in discussed in Section 5 and conclusions summarized in Section 6.

## 2 Study basins and data sets

Within a closed hydrologic basin, the annual water budget equation can be summarized as:

$$P - Q - E_T = dS/dt \tag{1}$$

where *P*, *Q* and $E_T$ [mm yr$^{-1}$] represent annual sums of fluxes, and *dS/dt* [mm yr$^{-1}$] is the annual change in terrestrial water storage. Besides *Q*, all other lateral water fluxes (into or out of the basin) are assumed to be negligible. See Section 2.2 below for a description of data products used to describe flux terms on the left-hand-

side of (1). The storage change term *dS/dt* is independently obtained using both gravity-based (GR) retrievals of total terrestrial water storage and passive microwave-based (PM) retrievals of surface soil moisture content. In both cases, annual change estimates are based on the differencing of temporally-averaged storage retrievals acquired at (or near) the end of each calendar year. Based on constraints associated with the availability of various remote sensing products, the analysis is conducted within a January 1, 2003 to December 31, 2010 time

period. Additional methodological details are given below.

### 2.1 Study basins

For the analysis, hydrologic basins are sought with: excellent ground-based rain gauge coverage, the availability of good remotely-sensed $E_T$ products, and the relative absence of complex topography and/or dense vegetation conditions known to reduce the accuracy of existing long-term, satellite-based soil moisture products. In addition,

arid areas are avoided due to their known lack of inter-annual *dS/dt* variations. The North American Mississippi River system is one of only a handful of continental-scale river basins which generally meets all of these criteria. Therefore, water budget closure will be examined in two separate sets of basins within the Mississippi River system. To start, a large-scale analysis will be conducted on five major Mississippi River sub-basins: the Missouri, the Arkansas, the Red, the Ohio and the Upper Mississippi - see Figure 1 and Table 1. The primary

focus in these large-scale basins will be evaluating the ability of existing *P*, *Q*, $E_T$ and GRACE-based *dS/dt* product to close the annual water budget. The results of this water balance analysis will subsequently be used to




refine the geographic focus and water flux processing approach applied in the medium-scale analysis described below.

Following this large-scale water balance analysis, the performance of a microwave-based *dS/dt* proxy is examined within 16 (smaller) medium-scale ($10^3$-$10^4$ km$^2$) unregulated basins positioned along an east/west

transect across the United States Southern Great Plains (SGP) region (see Figure 1 and Table 2). A complete justification of this geographic emphasis is given in Section 3. However, in general, medium-scale basins were selected following a screening analysis applied by the Model Parameter Estimation Experiment project (Duan et al., 2006) which removed basins with either inadequate rain gauge coverage or excessive human regulation of stream flow. Moving from west to east, these basins exhibit progressively higher mean *P* and annual runoff ratios

(*Q/P*) (Figure 1 and Table 2). Associated with this climatic gradient is a gradual west-to-east increase in vegetation biomass. Western basins are characterized by large fractions of rangeland, grassland and winter wheat land cover with relatively low biomass. In contrast, basins located along the eastern edge of the transect contain significant upland forest cover and intensive summer agricultural cultivation in low-lying areas.

### 2.2 Data Products and Processing

A range of ground and remotely-sensed data sets were acquired to characterize components of the terrestrial water balance summarized in (1). The acquisition and processing of these datasets is described below.

### 2.2.1 Thermal remote sensing of $E_T$

Daily evapotranspiration estimates were obtained from the Atmosphere-Land EXchange Inverse (ALEXI) algorithm. In particular, ALEXI exploits the moisture signal conveyed by the mid-morning rise in satellite

observed land surface temperature (LST) in order to capture water limitations on surface energy fluxes (Anderson et al. 2007a,b; Hain et al. 2009, 2011). Based on this principle, ALEXI produces estimates of daily evapotranspiration without any direct knowledge of antecedent precipitation or soil water balance considerations (Anderson et al., 2011). This ensures that ALEXI evapotranspiration estimates are independent of those derived via water balance calculations.

ALEXI evapotranspiration has been evaluated using a spatial disaggregation technique (DisALEXI) which uses high resolution LST retrievals from Landsat to downscale ALEXI fluxes to a 30-m pixel level (Anderson et al.,





2004). Typical accuracies obtained in comparison with eddy-covariance tower observations are on the order of 5 to 15% for daily to seasonal evapotranspiration estimates during snow-free periods (Anderson et al., 2012; Cammalleri et al., 2013, 2014a; Semmens et al., 2016).

Here, the ALEXI model was processed over CONUS at a spatial resolution of 4-km for the period of 2003-2010
and forced with: meteorological inputs from the Climate Forecast System Reanalysis (CFSR; Saha et al. 2010), TIR land surface temperature from the Geostationary Operational Environmental Satellites (GOES East and West), and leaf area index estimates obtained from the 4-day 1-km Combined Aqua-Terra MODIS product (MCD15A3).

Daily, instantaneous clear-sky latent heat fluxes retrieved from ALEXI were upscaled to daytime-integrated
evapotranspiration estimates assuming a self-preservation of the ratio of latent heat flux and incoming shortwave radiation ($f_{SUN}$) during daytime hours (Cammalleri et al., 2014b). Hourly CFSR incoming shortwave radiation inputs were integrated to produce daily estimates (24-h) of insolation used in this temporal upscaling. Currently, ALEXI is not executed over snow-covered surfaces. These periods were instead gap-filled with a linear interpolation of $f_{SUN}$ and a snow albedo correction to account for differences in surface net radiation over snow-
covered versus snow-free surfaces. Resulting 4-km ALEXI daily evapotranspiration estimates were temporally-summed within calendar years to produce annual $E_T$ [mm yr$^{-1}$] and spatially-averaged within each of the basins listed in Tables 1 and 2. Annual ALEXI $E_T$ estimates acquired in this way have been successfully applied to verify inter-annual evapotranspiration estimates acquired from land surface modeling (Han et al., 2015).

### 2.2.2 Land surface model predictions of $E_T$

For the purposes of cross-comparison with ALEXI $E_T$ results, annual $E_T$ was also acquired from 0.125°-resolution Noah v3.2 simulations (Chen et al., 1996; Chen and Dudhia, 2001; Ek et al., 2003) generated as part of North American Land Data Assimilation Phase 2 (NLDAS-2) activities (Xia et al. 2012). Hourly Noah predictions of: 1) direct evaporation from the surface soil, 2) direct evaporation of canopy-intercepted precipitation, and 3) transpiration via plant root uptake of water were aggregated to produce an hourly evapotranspiration estimate.
Annual $E_T$ averages were then obtained by summing these estimates for the calendar years 2003 to 2010 and spatially-averaging these summations within the basins indicated in Figure 1.





### 2.2.3 Ground-based observations of *P* and *Q*

Daily stream flow magnitudes were obtained from United States Geologic Survey (USGS) stream gauging stations located at the outlet of basins listed in Tables 1 and 2. These values were aggregated into (calendar year) sums and normalized by basin drainage area to obtain units of water depth per year [mm yr$^{-1}$]. Annual total

(liquid plus solid phase) precipitation ($P$) [mm yr$^{-1}$] was based on the temporal aggregation of rain gauge observations acquired by the National Centers for Environmental Prediction (NCEP)'s Climate Prediction Center (CPC) and re-sampled onto a 0.125° grid by the NLDSE-2 project (Xia et al., 2012). These annual averages were then spatially-averaged within each of basins listed in Tables 1 and 2.

### 2.2.4 Gravity remote sensing of *dS/dt*

Monthly GRACE terrestrial water storage deviations ($S_{GR}$) data were obtained by separately applying the rescaling coefficients of Landerer and Swenson (2012) to gridded 1° GRACE Level-3 terrestrial water storage products provided by the GeoForschungsZentrum (GFZ; version RL05.DSTvSCS1409), University of Texas Center for Space Research (CSR; version RL05.DSTvSCS1409), and the NASA/Cal-Tech Jet Propulsion Laboratory (JPL; version RL05.DSTvSCS1411). GRACE-based annual estimates of terrestrial water storage

variations ($dS_{GR}/dt$) were then derived via simple linear averaging of the GFZ, CSR and JPL terrestrial storage product to estimate $S_{GR,Dec,i}$ and $S_{GR,Jan,i+1}$ (where $i$ is an annual index) and the subsequent application of year-over-year differencing:

$$(dS_{GR}/dt)_{,i} = (S_{GR,Dec,i} + S_{GR,Jan,i+1})/2 - (S_{GR,Dec,i-1} + S_{GR,Jan,i})/2. \qquad (2)$$

Finally, gridded 1° $dS_{GR}/dt$ [mm yr$^{-1}$] products were spatially-averaged within all of the coarse-scale basins listed

in Table 1. Note that GRACE Level-3 $S_{GR}$ values capture monthly deviations from a long-term average datum (based on average 2004-2009 conditions) and not absolute $S$ values. However, the distinction is immaterial since our focus lies solely on annual $dS_{GR}/dt$, which is insensitive to the presence or absence of any such datum.

The primary application of $dS_{GR}/dt$ retrievals will be to verify annual water balance closure within the coarse-scale basins listed in Table 1. However, we will also apply $dS_{GR}/dt$ within the medium-scale basins as a source of

parameterization information for microwave-based $dS/dt$ estimates and as a baseline for evaluating microwave-based $dS/dt$ as a source of downscaling information (see Section 4.2). Naturally, these applications will be





approached with caution since the spatial resolution of the $dS_{GR}/dt$ retrievals (~200,000 km$^2$) is much coarser than the size of the medium-scale basins (2,000-10,000 km$^2$). The impacts of this significant scale mismatch will be discussed below.

### 2.2.5 Passive microwave remote sensing of soil moisture

Passive microwave-based surface soil moisture retrievals were based on the application of the Land Parameter Retrieval Model (LPRM; Owe et al., 2001) to Advanced Microwave Scanning Radiometer – EOS (AMSR-E) C- and X-band brightness temperature observations obtained from both ascending (1:30 PM LST) and descending (1:30 AM LST) orbits of the NASA Aqua satellite (Owe et al., 2008). AMSR-E LPRM Level 3 soil moisture data products were downloaded from the NASA Global Change Master Directory (http://gcmd.nasa.gov). The Aqua

satellite was launched in June 2002 and remained operational until October 2011. Soil moisture datasets acquired from AMSR-E represent the longest surface soil moisture data record currently available from a single satellite sensor. The processing of these datasets into $dS/dt$ estimates is discussed in Section 4.

### 2.3 Statistical approach

The temporal length of required remotely-sensed data sets imposes a serious challenge for this analysis. The

primary limiting factor for this length is the availability of a consistent microwave-based $\theta$ dataset. The data period utilized here (2003-2010) represents the longest current period of (temporally-consistent) microwave-based $\theta$ retrievals available from any single sensor (AMSR-E). Nevertheless, it still provides only eight annual values upon which to evaluate the annual closure of (1). Longer $\theta$ datasets based on the merger of multi-satellite/multi-sensor $\theta$ retrievals exist (Liu et al., 2011). However, concerns about their temporal consistency

currently limit their value for analyses conducted at inter-annual time scales (Loew et al., 2013).

The restriction of the annual analysis to only 8 years limits our ability to robustly assess closure using temporal sampling alone. Therefore, whenever possible, we will sample closure evaluation statistics across both space and time to maximize the total degrees of freedom available for a statistical analysis. However, due to significant amounts of both spatial and temporal auto-correlation in $P$-$Q$-$E_T$ datasets, considerations must be made for

oversampling (in both space and time) when calculating effective sample sizes. To address this we followed the approach of Bretherton et al. (1999) who recommended (for the case of sampling quadratic statistics) an effective sampling size $N^*$ of:



$$N^* = N(1 - \rho^2)/(1 + \rho^2) \qquad\qquad (3)$$

where $N$ is the original sample size and $\rho$ the auto-correlation at individual sampling points. In particular, we applied (3) separately in both space and time utilizing both temporal (separated in time, sampled over time and then averaged across various basins) and spatial (separated in space, sampled over space and then averaged over

various years) samples of $\rho$ to obtain both spatial and temporal sample size reduction factors. Next, the total sample size (i.e., total time samples x total space samples) was multiplied by both reduction factors to estimate effective sample size in both time and space. For example, in the large-scale basin analysis, we have a total sample size of 40 annual values (5 basins over 8 years); however, after accounting for over-sampling in both space and time, the effective sample size was reduced to 9.7. Likewise, for the medium-scale basins analysis, the

total sample size of 128 annual values (16 basins over 8 years) was reduced to an actual effective size of 48.4. These effective sample sizes were then used to calculate effective degrees-of-freedom for all statistical hypothesis tests.

**3 Water balance closure within large-scale basins**

All water storage and flux products described above contain significant errors and biases. In addition, it is

possible that non-resolved flux terms in (1) may hinder closure versus observed storage changes. Therefore, before deriving and evaluating an approach to estimate $dS/dt$ for medium-scale basins using microwave-based remote sensing, we will first verify the ability of water balance data sets introduced in Section 2 to close the terrestrial water balance via (1). Due to the coarse spatial resolution of GRACE, a direct closure analysis is possible only for the large-scale basins listed in Table 1. Based on $E_T$ values derived from ALEXI, Figure 2 plots

annual variations of $P$-$Q$-$E_T$ and (GRACE-based) $dS_{GR}/dt$ for all 5 large-scale basins listed in Table 1. In all basins except the Missouri, annual values of $P$-$Q$-$E_T$ depart significantly from zero – illustrating the general importance of annual $dS/dt$ on the terrestrial water budget. Within the Missouri, $P$-$Q$ is roughly balanced by $E_T$, and therefore, alone among other basins examined here, the annual estimation of $dS_{GR}/dt$ does not appear to be a requirement for closing the annual water budget. This may be linked to the very large reservoir capacity of the

Missouri River Basin system, and the active management of $Q$ to minimize inter-annual reservoir and channel level variability. This aggressive level of management ensures that the Missouri River Basin exhibits the smallest standard deviation of inter-annual $P$-$Q$-$E_T$ variability (~30 mm yr$^{-1}$- see Figure 2) of any large basin considered in this analysis.





The best closure results in Figure 2 are obtained in the Arkansas River and Red River basins. In these two basins, GRACE-based $dS_{GR}/dt$ closely matches inter-annual variations in $P-Q-E_T$. This suggests that in the United States Southern Great Plains (SGP) region, both the assumptions underlying (1) and the water flux data sets considered are sufficiently accurate to characterize inter-annual variations in $S$. In contrast, there is clear evidence of a low

bias in annual $P-Q-E_T$ relative to $dS_{GR}/dt$ within both the Upper Mississippi and Ohio River Basins. Given the frequency and extent of winter-time snow cover in these basins, it seems reasonable to ascribe this bias to known under-catch issues associated with the gauge-based measurement of snowfall (Goodison et al., 1998). In addition, there exists a potential for systematic error in cold-season ALEXI $E_T$ estimates (which are based on a simple extrapolation approach).

Figure 3a show annual $P-Q-E_T$ versus $dS_{GR}/dt$ for all 5 large-scale basins. The sampled correlation is in Figure 3a is marginal (0.37 [-]) but improves considerably (0.52 [-]) when the 8-year mean of annual $P-Q-E_T$ is subtracted from yearly $P-Q-E_T$ results for each basin (Figure 3b). This is equivalent to imposing closure of $P-Q-E_T$ within each basin over the entire 8-year time period. In addition, replacing ALEXI $E_T$ with Noah-based $E_T$ reduces the sampled correlations in both Figure 3a and 3b (from 0.37 to 0.33 [-] and from 0.52 to 0.33 [-], respectively). This

implies that preference should be given to the remotely-sensing-based ALEXI $E_T$ product.

Due to the coarse spatial resolution of GRACE-based $dS_{GR}/dt$, a comparable water balance analysis cannot be applied to the medium-scale basins listed in Figure 1 and Table 2. Instead we will cross-apply general tendencies observed in the large-scale closure analysis (Figures 2 and 3) to refine the medium-scale analysis presented below. In particular, the medium-scale basins listed in Table 2 are selected based on the principal of minimization

of both human regulation (to avoid the lack of annual $P-Q-E_T$ signal noted in the Missouri Basin) and snow/could season impacts (to avoid the low bias in annual $P-Q-E_T$ observed in the Ohio and Upper Mississippi River Basins). Overall, these two considerations motivate our decision to utilize only lightly-regulated MOPEX basins within the SGP portion of the Mississippi River system (see Figure 1 and earlier discussion in Section 2.1). In addition, based on annual water balance closure results presented in Figures 2-3, ALEXI-based (as opposed to

Noah-based) $E_T$ will be used and closure will be imposed on 8-year $P-Q-E_T$ totals.



**4 Microwave-based closure for medium-scale basins**

As discussed above, the primary focus of the paper is on resolving inter-annual variations of $P$-$Q$-$E_T$ for a series of medium-scale basins using a new microwave-based proxy for $dS/dt$. This section will describe the derivation of the proxy and its empirical evaluation within the medium-scale basins listed in Table 2.

**4.1 Microwave-based $dS/dt$ estimation**

Any transition between surface soil moisture and $S$ must account for relative variations in the temporal scale and phase of both quantities. In particular, the tendency for $S$ variations is temporally-smoothed, and lagged (in time), with respect to corresponding surface soil moisture variability (Chagnon 1987; Entekhabi et al., 1992; Swenson et al., 2008). Based on this reasoning, instantaneous 0.25° LPRM surface soil moisture retrievals (see Section 2.2)

were averaged in time and space into a single monthly value for each of the basins in Tables 1 and 2. Next monthly (basin-scale) soil moisture averages for October, November and December ($\theta_{PM,Oct}$, $\theta_{PM,Nov}$, and $\theta_{PM,Dec}$) were merged into a single, end-of-calendar-year estimate of passive-microwave based $\theta_{PM}$:

$$\theta_{PM,i} = W_{Oct}\,\theta_{PM,Oct,i} + W_{Nov}\,\theta_{PM,Nov,i} + W_{Dec}\,\theta_{PM,Dec,i} \qquad (4)$$

where $i$ is an annual index (here corresponding to calendar years between 2003 and 2010), and $W$ are constant

weighting factors (summing to unity) applied to each month. Annual changes in $\theta_{PM}$ ($d\theta_{PM}/dt$) were then derived from annual differencing of $\theta_{PM,i}$ with $\theta_{PM,i-1}$. This entire procedure was done separately for LPRM retrievals acquired during both ascending and descending AMSR-E orbits. Finalized values of $d\theta_{PM}/dt$ were then obtained by averaging results obtained from both orbits. Our decision to utilize a calendar year to accumulate annual flux/storage change totals in (1) is largely arbitrary, and the impact of utilizing other annual periods will be

discussed below.

In addition to the specification of $W$, we also allowed for the application of a single calibration factor $K_{PM}$ [mm] when converting volumetric $d\theta_{PM}/dt$ [$m^3 m^{-3}$ $yr^{-1}$] variations into annual $dS/dt$ depth changes [mm $yr^{-1}$]:

$$dS_{PM}/dt = K_{PM}\,d\theta_{PM}/dt. \qquad (5)$$

Our approach for obtaining $K_{PM}$ was based on scaling $d\theta_{PM}/dt$ to match the sampled temporal variance of gravity-

based $dS_{GR}/dt$. Therefore, $K_{PM}$ was defined as the ratio:





$$K_{PM} = \sigma(dS_{GR}/dt)/\sigma(d\theta_{PM}/dt) \tag{6}$$

where the $\sigma$ operator indicates a temporally-sampled inter-annual standard deviation.

Despite some evidence for significant large-scale correlation between $\theta$ and $S$ (Abelen et al., 2013; 2015), there are strong *a priori* reasons for scepticism regarding the application of (4-6) to a water budget application. First,
due to the extremely shallow vertical support of passive microwave-based surface soil moisture retrievals, it is uncertain if $d\theta_{PM}/dt$ actually provides an effective linear proxy for $dS/dt$. Second, even if such a linear relationship can be established, it is unclear if the ratio $\sigma(dS_{GR}/dt)/\sigma(d\theta_{PM}/dt)$ in (6) provides a robust calibration coefficient to convert surface soil moisture variations into annual variations in $S$. Below we will attempt to provide empirical evidence to allay these (credible) theoretical concerns.

**4.2 Evaluation of proxy assumptions and calibration**

Figure 4 plots (annual) variations of $P$-$Q$-$E_T$ and $dS_{PM}/dt$ for all 16 medium-scale basins listed in Table 1. See Section 3 for the rationale behind the selection of these particular basins. The large plotted departures (from zero) seen for $P$-$Q$-$E_T$ confirms that inter-annual variations in $S$ play a significant role in the application of (1) at an annual time scale.

In addition, consistently negative $P$-$Q$-$E_T$ estimates are observed within several medium-scale basins (see e.g., basins #5, #8, #9, and #12 in Figure 4). Because these basins cannot be directly resolved by GRACE, it is difficult to confirm whether this tendency is a real (i.e., a decadal scale reduction in $S$) or an artefact of the summed impact of multiple long-term measurement biases in independent $P$, $Q$ and $E_T$ products. However, based on the large-basin analysis presented in Section 3, the latter appears more likely. Therefore, annual $P$-$Q$-$E_T$ values
are de-biased by subtracting out (on a basin-by-basin basis) the 8-year annual mean of $P$-$Q$-$E_T$ (see dashed line in Figure 4). The impact of this assumption on subsequent results will be discussed below.

Our primary goal is determining the potential for explaining observed annual $P$-$Q$-$E_T$ variations in Figure 4 using the microwave-based $dS_{PM}/dt$ proxy introduced above. Our first priority is empirically evaluating the assumptions - expressed in (4-6) - which underlie the proxy. The first issue is the degree to which the appropriate temporal
averaging of microwave-based soil moisture via (4) can be used to obtain a robust linear proxy for $P$-$Q$-$E_T$. Figure 5a addresses this by plotting the average linear correlation for all the medium-scale basins between annual





$P$-$Q$-$E_T$ and $d\theta_{PM}/dt$ obtained using all potential combinations of $W_{Dec}$, $W_{Nov}$ and $W_{Oct}$ (where $W_{Dec} + W_{Nov} + W_{Oct} =$ 1.0). Plotted correlations in Figure 5a are generally greater than 0.50 [-]. In fact, even after realistically accounting for the impact of over-sampling due to spatial and temporal auto-correlation in the $P$-$Q$-$E_T$ fields (Section 2.3), sampled correlations are statistically-significant (one-tailed, 95% confidence) for all possible

combinations of $W_{Dec}$, $W_{Nov}$ and $W_{Oct}$. Since these correlations are based on annual values (where there is no potential for obtaining spurious fitting due to the trivial representation of an obvious seasonal cycle), and there is no credible reason to suspect cross-correlated errors between the wholly independent $P$-$Q$-$E_T$ and $dS_{PM}/dt$ fields, the statistical significance of sampled correlation in Figure 5a can be taken as clear evidence of a linear underlying relationship between $d\theta_{PM}/dt$ and $P$-$Q$-$E_T$. As such, it provides empirical support for (4-5).

Nevertheless, the performance of the $d\theta_{PM}/dt$ proxy does vary as a function of $W_{Dec}$, $W_{Nov}$ and $W_{Oct}$ in Figure 5a and feasible parameterization strategies will be required to fix their values. To this end, Figure 5b plots the sampled correlation between $d\theta_{PM}/dt$ and $dS_{GR}/dt$ as a function of $W_{Dec}$, $W_{Nov}$ and $W_{Oct}$. Note that monthly weighting values which maximize this correlation in Figure 5b also tend to maximize the correlation between $d\theta_{PM}/dt$ and $P$-$Q$-$E_T$ in Figure 5a. Based on Figure 5b, the maximum correlation between $d\theta_{PM}/dt$ and $dS_{GR}/dt$ is

found at $W_{Oct} = 0.4$ [-], $W_{Nov} = 0.5$ [-], and $W_{Dec} = 0.1$ [-]. These (spatially and seasonally-fixed) weighting values will be used for all subsequent calculations of $d\theta_{PM}/dt$ via (4). The relative lack of weight applied to December surface soil moisture retrievals is likely reflective of frozen soil moisture conditions at this time and the need for $S$ anomalies to be lagged in time with respect to superficial surface soil moisture variations.

This parameterization of $W_{Oct}$, $W_{Nov}$, and $W_{Dec}$ is sufficient if $d\theta_{PM}/dt$ is to be interpreted solely as a linear proxy

for relative inter-annual variations in $dS/dt$; however, interpretation of $d\theta_{PM}/dt$ as an absolute measure will require the additional parameterization of $K_{PM}$ [mm] in (5) to transform $d\theta_{PM}/dt$ into a representation of $dS/dt$ with units of [mm yr$^{-1}$] (i.e., $dS_{PM}/dt$). Figure 6 shows the impact of $K_{PM}$ in (5) on the root-mean-square difference (RMSD) between $dS_{PM}/dt$ and $P$-$Q$-$E_T$. Results are obtained by lumping annual results for all years within all medium-scale basins listed in Table 2 and the assumption that $K_{PM}$ is fixed in both space and time. The plotted horizontal

line plots the inter-annual standard deviation of $P$-$Q$-$E_T$ - which is equivalent to the RMSD accuracy achievable by assuming $dS/dt = 0$ in (1). This baseline is improved upon by a wide range of $K_{PM}$ values. However, the absolute accuracy of the $dS_{PM}/dt$ proxy is maximized near $K_{PM} = 900$ mm.





The $K_{PM}$ estimation approach in (6) is based on the assumption that this optimal value can be obtained via a simple variance matching approach applied to $d\theta_{PM}/dt$ and $dS_{PM}/dt$. Applying (6) (in a lumped manner to all years and all medium-scale basins in Table 2) leads to a value of $K_{PM} = 1080$ mm, which is reasonably close to the optimal value of $K_{PM}$ (900 mm). It is also well-within the broad range of $K_{PM}$ which improves upon a baseline of

neglecting $dS/dt$ entirely (see Figure 6).

It should be noted that the parameterization strategies presented above involve direct comparison between (relatively) high-resolution $\theta$ products obtained from microwave remote sensing with lower-resolution GRACE-based $dS_{GR}/dt$ retrievals (which have been trivially re-sampled to capture a basin-scale mean). Despite the inability of GRACE retrievals to spatially-resolve the medium-scale basins considered here, Figures 5 and 6

suggest these comparisons are still able to yield useful calibration information. However, it is possible that resolution inconsistencies between GRACE and AMSR-E may have a degrading impact on results. One strategy for resolving this scale inconsistency is to first degrade the spatial resolution of the AMSR-E $\theta$ field to match the ~200,000 km$^2$ GRACE resolution prior to applying the calibration approach outlined in Figures 5a and 6. However, attempts to do this (via smoothing of the AMSR-E $\theta$ fields using a 2-dimensional Gaussian filter)

actually led to a small *decrease* in the quality of the $W_{Oct}$, $W_{Nov}$, $W_{Dec}$, and $K_{PM}$ calibration. This implies that, despite their resolution differences, direct comparisons between AMSR-E and GRACE products appears to offer the most viable calibration approach.

**4.3 Microwave-based closure evaluation**

Utilizing the calibrated $W$ and $K_{PM}$ derived in Section 4.2 leads to the $dS_{PM}/dt$ values plotted in Figure 7. Each

point in the scatter plot represents one annual value within a single medium-scale basin. Our microwave-based $dS_{PM}/dt$ proxy product has a linear correlation with independently-acquired $P$-$Q$-$E_T$ values of 0.71 [-], which is statistically-significant (one-tailed, at 99% confidence) even after allowances have been made for over-sampling in both time and space (see Section 2.3). Note that all calibrated parameters ($W$ and $K_{PM}$) are constant in both space and time and therefore cannot provide a spurious source of skill for $dS_{PM}/dt$ temporal variations. In

addition, all calibration is against GRACE-based $dS_{GR}/dt$ and plotted $P$-$Q$-$E_T$ values are used solely for the purpose of independent verification.

While $P$-$Q$-$E_T$ derived in medium-scale basins cannot be directly validated against GRACE-based $dS_{GR}/dt$ retrievals (due to the ground-resolution of GRACE being much coarser than the size of the medium-scale basins),





the significant correlation in Figure 7 strongly suggests that they are adequately representing the net annual flux of water into and out of the medium-scale basins. A slight reduction in correlation (from 0.71 to 0.62 [-]) is found when $P$-$Q$-$E_T$ is *not* corrected to close water balance over the 8-year study period. Likewise, replacing ALEXI $E_T$ with NOAH-based $E_T$ leads to another (very) slight reduction in correlation in Figure 7 (from 0.71 to 0.69 [-]).

However, it should be stressed that, in all cases, the correlation between $dS_{GR}/dt$ and plotted $P$-$Q$-$E_T$ remains statistically significant (one-tailed, at 95% confidence). See Figure 4 for $dS_{PM}/dt$ time series results within individual medium-scale basins.

### 4.4 Downscaling evaluation

An important follow-on question is the degree to which the skill demonstrated in Figure 7 enhances our ability to

track $dS/dt$ in medium-scale basins above and beyond existing GRACE products. To this end, Figure 8a plots annual GRACE-based $dS_{GR}/dt$ versus $P$-$Q$-$E_T$ for all medium-scale basins. Since the ground resolution of GRACE is significantly coarser than the size of these basins, it is unfair to evaluate $dS_{GR}/dt$ based on these comparisons. However, despite this severe resolution penalty, $dS_{GR}/dt$ still manages to correlate relatively well (i.e., a linear correlation of 0.66 [-]) with independently-acquired estimates of annual $P$-$Q$-$E_T$. The tendency for

skill in GRACE-based $dS_{GR}/dt$ to persist even at these (sub-resolution) scales implies that annual $dS/dt$ fields in this region contain spatial auto-correlation at length scales finer than the GRACE spatial resolution. However, it should be stressed that the use of GRACE-based $dS_{GR}/dt$ fields at these spatial resolutions is not recommended and applied here only to define a baseline upon which to evaluate the benefits of subsequent downscaling using microwave-based $dS_{PM}/dt$ estimates.

To this end, Figure 8b plots the relationship between annual $P$-$Q$-$E_T$ and $dS/dt$ estimates obtained via a simple downscaling strategy based on the direct averaging of annual $dS_{GR}/dt$ and $dS_{PM}/dt$ estimates for each medium-scale basin. Relative to GRACE-only results presented in Figure 8a, this simple downscaling strategy leads to a significant improvement in the degree of correlation with independent $P$-$Q$-$E_T$ values. Specifically, this correlation is increased from 0.66 [-] for the GRACE-only $dS_{GR}/dt$ case in Figure 8a to 0.77 [-] for the case of

averaging $dS_{GR}/dt$ and $dS_{PM}/dt$ in Figure 8b. Application of a Fisher $z$-transformation and the effective degree sample size calculation presented in Section 2.3 confirms that this increase in correlation is statistically significant (two-tailed, at 95% confidence).





In order to further examine geographic trends in results, Figure 9 evaluates $dS_{PM}/dt$, $dS_{GR}/dt$ and downscaling results (based on the simple linear averaging $dS_{PM}/dt$ and $dS_{GR}/dt$) obtained individually for each medium-scale basin in Table 2. Results are shown for both the linear correlation and absolute RMSD match with annual $P$-$Q$-$E_T$ variations. It is reasonable to expect that the accuracy of microwave-based $\theta$ retrievals, and thus their value as the basis of $dS_{PM}/dt$ estimates, should progressively degrade for higher-numbered study basins (which generally have wetter climates and denser vegetation coverage – see Figure 1). Therefore, it is somewhat surprising that no clear trend between basin land cover and the relative performance of the microwave based $dS_{PM}/dt$ proxy is discernible in Figure 9. However, $dS_{PM}/dt$ results demonstrate relatively poor accuracy for the furthest north (and most heavily-cultivated) basin (i.e., basin #7) and for the wettest basin (i.e., basin #16). The downscaled results (based on the simple averaging of $dS_{PM}/dt$ and $dS_{GR}/dt$) generally produce results which are superior to the isolated application of either $dS_{PM}/dt$ or $dS_{GR}/dt$; however, basin-to-basin variations are large and metric values for individual basins are likely to be impacted by large sampling errors.

It is possible to replicate the $dS_{PM}/dt$ approach applied to the medium-scale basins for the larger-scale basins listed in Table 1. However, large-scale $dS_{PM}/dt$ proxies calculated in this way (not shown) are significantly less accurate than GRACE-based $dS_{GR}/dt$ results. There is no indication that a microwave-based $dS_{PM}/dt$ proxy can consistently improve upon the relative accuracy of annual $dS/dt$ in large basins beyond what is already possible via the utilization of existing GRACE-based $dS_{GR}/dt$. As a result, the added benefits of a microwave-based $dS_{PM}/dt$ proxy appear limited to basins which cannot be directly resolved by GRACE.

## 5 Discussion

Passive microwave-based estimates of surface soil moisture capture only soil water storage variations occurring within the couple of centimetres of the vertical soil column and cannot directly detect storage dynamics occurring in deeper layers of the unsaturated zone - to say nothing of even deeper variations in groundwater storage or reservoir storage. However, despite this severe theoretical limitation, passive microwave surface soil moisture retrievals ($\theta$) appear to retain significant value as an indicator of relative inter-annual variations in $P$-$Q$-$E_T$ (see e.g., Figure 7). This implies that, at least at an annual time scale and for certain conditions, unobserved components of $S$ are sufficiently correlated with *observable* trends in surface soil moisture such that $\theta$ retrievals may serve as a potential proxy for variations in total $S$. Given the two orders of magnitude difference in the spatial resolution of microwave-based $\theta$ (1,000 km$^2$) versus gravity-based (200,000 km$^2$) $dS/dt$ estimates, the





microwave-based proxy appear to enhance our existing ability to closure the terrestrial water budget within the medium scale (2,000-10,000 km$^2$) basins listed in Table 2 (Figure 8).

Intuitively, the ability of surface $\theta$ retrievals to capture (much deeper) $S$ variations is likely due to the tendency for (non-anthropogenic) variations in $dS/dt$ to be forced, in a "top-down manner", by atmospherically-driven
anomalies in $P$ and $E_T$. In this simple conceptual model, variations in surface soil moisture provide a leading indicator of these anomalies as they are propagated downward into deeper hydrologic storage units (Chagnon 1987; Entekhabi et al., 1992; Swenson et al., 2008). However, it must be stressed that this conceptual model is likely to break down for a number of cases; in particular, in instances in which variations in $S$ are forced by anthropogenic modification of the hydrologic cycle. For example, $S$ variations due to direct ground-water
pumping (Rodell et al., 2009), particularly when associated with increased surface soil moisture via irrigation, will almost certainly confound the ability of $\theta$ retrievals to effectively capture inter-annual variability in $S$. Likewise, it is difficult to imagine microwave-based $\theta$ providing an effective representation of variations in $S$ due to large changes in reservoir storage and/or river system management. Finally, even in cases lacking significant anthropogenic modifications of the hydrologic cycle, the relationship between soil moisture and groundwater
memory is known to vary significantly as a function of climate (Lo and Famiglietti, 2010). Some modes of soil moisture/groundwater interactions are almost certainly inconsistent with the application of (4-6). Therefore, additional study is required to better understand the geographic limitations of $d\theta_{PM}/dt$ as a credible $dS/dt$ proxy.

The geographic scope of this study was limited by two considerations. First, the evaluation analysis required access to sufficiently accurate annual $P$-$Q$-$E_T$ time series to serve as an independent benchmark for microwave-
based $dS_{PM}/dt$ estimates. As discussed in Section 2, this requirement restricts the geographic domain over which the analysis can currently be conducted. Second, the long-term AMSR-E LPRM soil moisture dataset utilized in the analysis has known limitations within areas of moderate and/or dense vegetation cover. Datasets based on lower-frequency L-band observations are currently being produced but will require 2 or 3 more years (beyond 2017) to match the temporal length of the existing AMSR-E data record. However, once longer-term L-band
datasets becomes available, they will enable the expansion of this analysis into more densely vegetated areas.

Our decision to calculate annual flux quantities using a calendar year (i.e., January 1 to December 31) approach is admittedly arbitrary. This choice will almost certainly impact the accuracy of $dS_{PM}/dt$ proxy estimates due to seasonal variations in the availability and accuracy of remotely-sensed soil moisture retrievals acquired from





passive microwave remote sensing (due to e.g. vegetation phenology and/or the presence of snow or frozen soils). The impact of frozen soils could, for example, be circumvented by defining years as ending in early fall and therefore requiring sampling of $\theta_{PM}$ only during spring and summer months. However, high amounts of vegetation biomass during these months leads to a higher amount of uncertainty in sampled $\theta_{PM}$ and thus $dS_{PM}/dt$.

A preliminary sensitivity analysis suggests that, despite the complication of frozen soil and snow cover, $dS_{PM}/dt$ results based on November to December $\theta_{PM}$ samples provide superior water budget closure than comparable results based on June to September $\theta_{PM}$. As a result, for the specific set of basins examined here, the use of a calendar year (January 1 to December 31) appears to maximize the value of $dS_{PM}/dt$ for water balance applications.

Finally, a natural extension of this work is the application of the $dS_{PM}/dt$ at a monthly (as opposed to annual) time scale. In theory this is possible; however, there are several practical obstacles which must be overcome. First, as noted above, the accuracy of the $dS_{PM}/dt$ proxy appears to be reduced when applied during heavier biomass conditions found outside of winter. This implies that it may be difficult to adequately characterize monthly-scale storage variations based on calculating $dS_{PM}/dt$ at multiple points over the season cycle. In addition, based on a

preliminary analysis, optimal values of $W$ and $K_{PM}$ appear to vary within the seasonal cycle. Therefore, a seasonally-varying parameterization would likely be required for $dS_{PM}/dt$ to accurately capture monthly variations in $S$. Given that monthly $dS_{PM}/dt$ variations are commonly dominated by a fixed seasonal cycle, it is very difficult to discern whether any apparent skill in monthly $dS_{PM}/dt$ variations is real or simply an artefact of over-fitting a seasonally-varying $W$ and/or $K_{PM}$ parameterization. As a result, the validation of a monthly $dS_{PM}/dt$ proxy will

likely require the availability of longer-term (i.e., 10+ years) $dS_{PM}/dt$ and $P$-$Q$-$E_T$ datasets capable of supporting mutually-exclusive calibration and validation time periods. As discussed above, the current limiting factor on the length of this analysis is the availability of temporally-consistent, satellite-based soil moisture products.

## 6 Conclusions

Advances in the remote sensing of $E_T$ currently afford an opportunity to independently verify other annual components of the terrestrial water budget - including changes in terrestrial water storage ($dS/dt$). Confirming recent work with GRACE, results clearly demonstrate the importance of $dS/dt$ for closing the annual water




budget. In particular, GRACE-based $dS_{GR}/dt$ estimates appear to provide a reliable source of such information within large-scale river basins with relative low annual snowfall totals and anthropogenic management (Figures 2-3). In addition, for basins smaller than the 200,000 km$^2$ GRACE spatial resolution, estimates of $dS_{PM}/dt$ derived from passive microwave remote sensing and (4-6) also demonstrate clear value for providing annual closure

information (Figure 7). Given that passive microwave-based soil moisture retrievals are available at substantially-finer spatial resolution than gravity-based retrievals of $S$, this opens up the strong possibility of using microwave-based surface soil moisture retrievals to downscale gravity-based $dS/dt$ retrievals (Figure 8).

The retrieval of the microwave-based $dS_{PM}/dt$ proxy is based on two - somewhat *ad hoc* - assumptions expressed in (4-6) which claim that: 1) $d\theta_{PM}/dt$ obtained via (4) has a linear underlying relationship with $dS/dt$ and 2) the

constant of proportionality in the relationship can be derived via variance matching between microwave and gravity-based estimates of $dS/dt$. These assumptions are directly supported by empirical results presented in Figures 5 and 6. Nevertheless, it should be stressed that theoretical support for (4-6) is still quite weak, and it is relatively easy to imagine cases in which these assumptions would be expected to fail (see Section 5). Therefore, additional validation work over a wider variety of conditions is certainly warranted.

In addition to isolating potential value in microwave-based $dS_{PM}/dt$ estimates, water balance results presented here also provide added confidence regarding our ability to capture annual variations in $dS/dt$ via (1) and flux observations. In particular, both annual $dS_{GR}/dt$ and $dS_{PM}/dt$ estimates exhibit a statistically-significant correlation against independent annual $P$-$Q$-$E_T$ values with the medium-scale basins examined here (Figure 7). Terrestrial $E_T$, in particular, is commonly perceived to represent a weak link in the characterization of (1). However, based on

results presented here, it appears that ALEXI-based $E_T$ products over CONUS are now sufficiently accurate (at least in a relative inter-annual sense) for annual $E_T$ estimates to be used as a viable constraint to infer the accuracy of other water budget components. This is a marked improvement over the calculation of $E_T$ as a balance residual and opens the door to the fuller use of (1) as a diagnostic tool for various water balance products.

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





**Table 1.** Attributes of large-scale basins in Figure 1.

| River Basin | USGS Station No. | USGS Station Name | Basin Size (km$^2$) | Annual P (mm) | Runoff Ratio Q/P |
|---|---|---|---|---|---|
| Missouri | 06934500 | Missouri River at Hermann, MO | 1,347,556 | 563 | 0.10 |
| Arkansas | 07263450 | Arkansas River at Little Rock, AR | 409,201 | 747 | 0.14 |
| Red | 07344370 | Red River at Spring Bank, AR | 153,906 | 850 | 0.13 |
| Upper Miss. | 07022000 | Mississippi River at Thebes, IL | 496,016 | 898 | 0.31 |
| Ohio | 03611500 | Ohio River at Metropolis, IL | 527,557 | 1187 | 0.45 |





**Table 2.** Attributes of medium-scale basins in Figure 1.

| Basin Number | USGS Station No. | USGS Station Name | Basin Size (km$^2$) | Annual P (mm) | Runoff Ratio Q/P |
|---|---|---|---|---|---|
| 1 | 07144780 | Ninnescah River AB Cheney Re, KS | 2,049 | 768 | 0.08 |
| 2 | 07144200 | Arkansas River at Valley Center, KS | 3,402 | 842 | 0.11 |
| 3 | 07152000 | Chikaskia River near Blackwell, OK | 4,891 | 896 | 0.19 |
| 4 | 07243500 | Deep Fork near Beggs, OK | 5,210 | 945 | 0.15 |
| 5 | 07147800 | Walnut River at Winfield, KS | 4,855 | 980 | 0.31 |
| 6 | 07177500 | Bird Creek Near Sperry, OK | 2,360 | 1025 | 0.23 |
| 7 | 06908000 | Blackwater River at Blue Lick, MS | 2,924 | 1140 | 0.29 |
| 8 | 07196500 | Illinois River near Tahlequah, OK | 2,492 | 1175 | 0.29 |
| 9 | 07019000 | Meramec River near Eureka, MO | 9,766 | 1187 | 0.28 |
| 10 | 07052500 | James River at Galena, MO | 2,568 | 1255 | 0.31 |
| 11 | 07186000 | Spring River near Wace, MO | 2,980 | 1258 | 0.27 |
| 12 | 07056000 | Buffalo River near St. Joe, AR | 2,148 | 1238 | 0.37 |
| 13 | 06933500 | Gascondade River at Jerome, MO | 7,356 | 1293 | 0.24 |
| 14 | 07067000 | Current River at Van Buren, MO | 4,351 | 1309 | 0.31 |
| 15 | 07068000 | Current River at Doniphan, MO | 5,323 | 1314 | 0.36 |
| 16 | 07290000 | Big Black River NR Bovina, MS | 7,227 | 1368 | 0.37 |





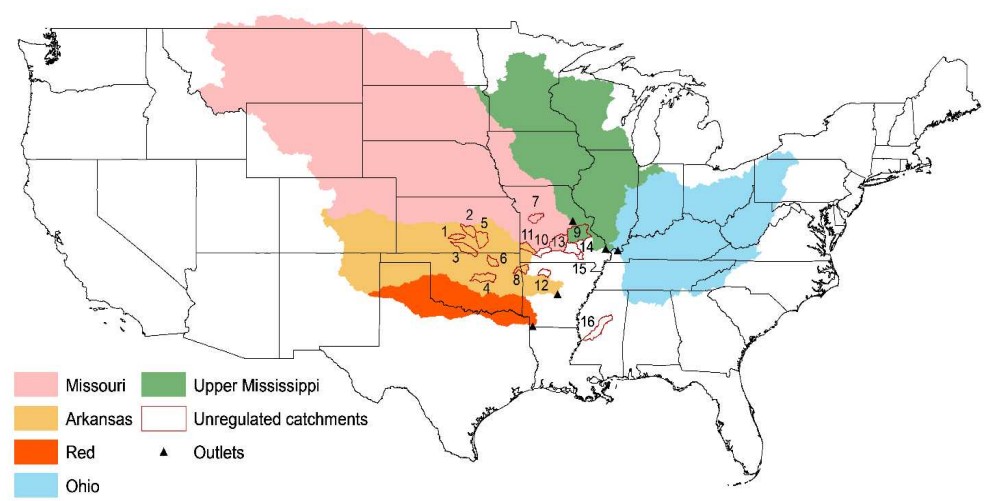

**Figure 1.** Map of the 5 large-scale basins (color shading - see Table 1) and 16 unregulated medium-scale basins (red outlines - see Table 2) considered in the analysis.





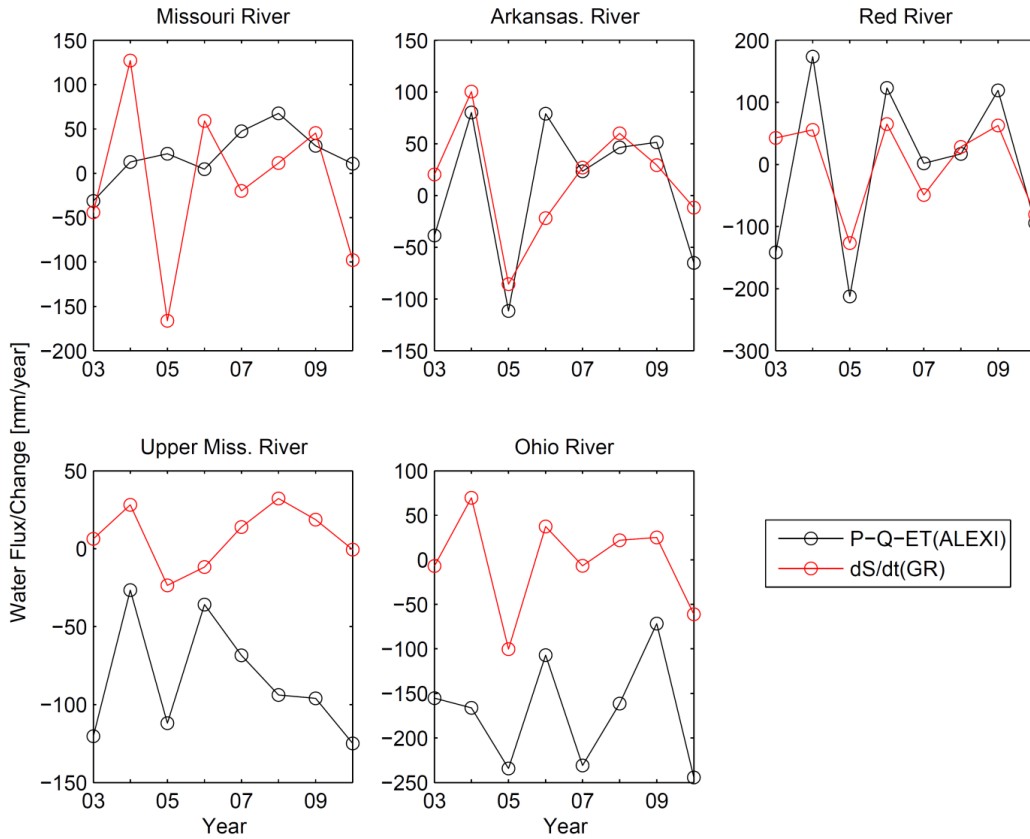

5   **Figure 2.** Annual time series of *P-Q-E$_T$* (black) and gravity-based *dS$_{GR}$/dt* (red) estimates for each of the large-scale basins listed in Table 1.





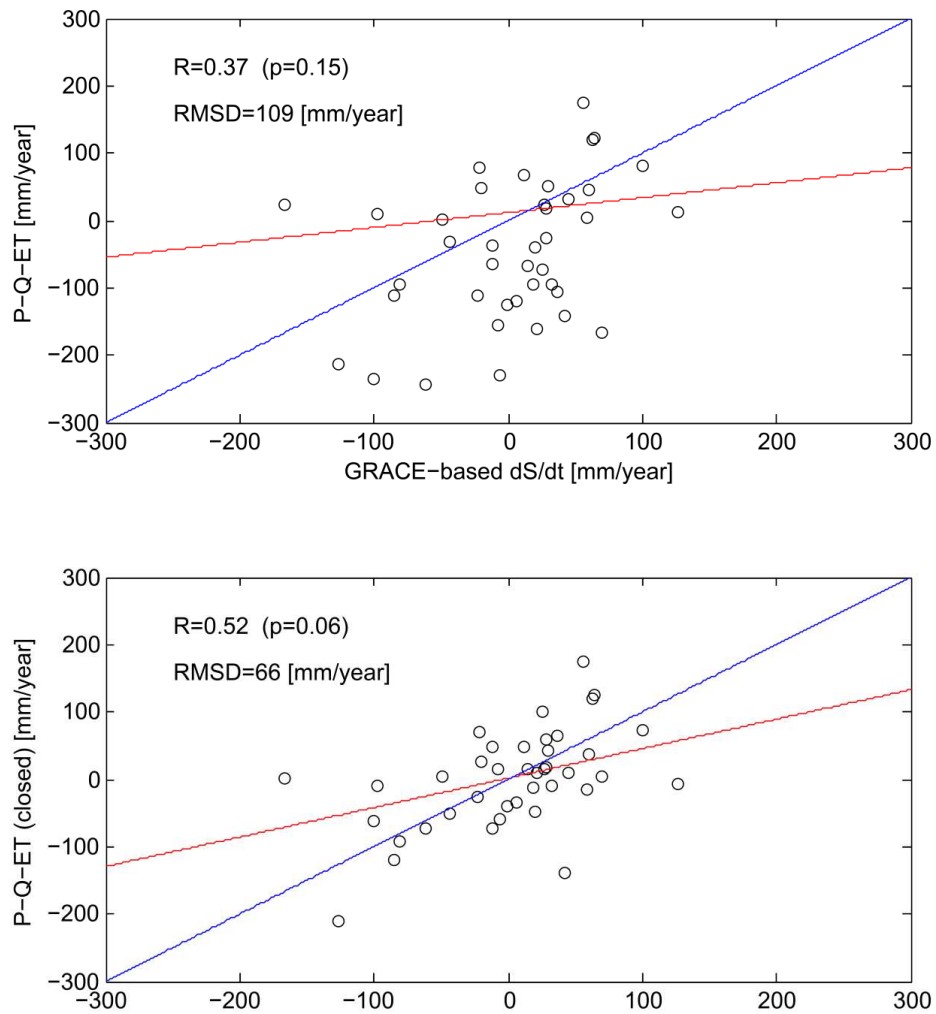

**Figure 3.** (a) Relationship between annual $P$-$Q$-$E_T$ and gravity-based $dS_{GR}/dt$ within each of the large-scale basins listed in Table 1. (b) Same, except that annual $P$-$Q$-$E_T$ time series for each basin have been closed (i.e., modified

5  to sum to zero over the 8-year data record). The blue line is a one-to-one line and red line is the least-squares linear fit.





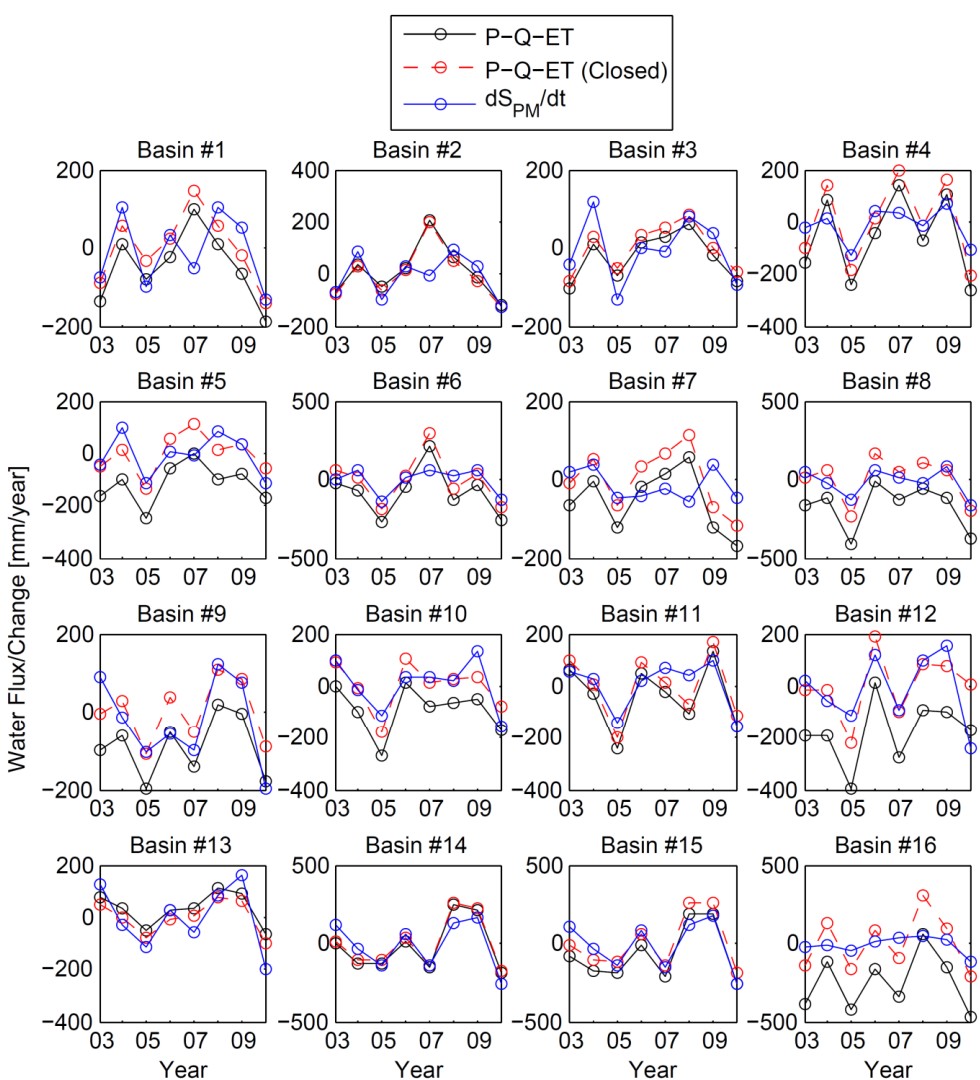

**Figure 4.** For the 16 medium-scale basins listed in Table 2, the annual time series of raw $P$-$Q$-$E_T$ (solid black line) and $P$-$Q$-$E_T$ obtained by assuming flux closure over the 8-year period of record (dashed red line). Values of the microwave-based $dS_{PM}/dt$ proxy are also plotted (solid blue line).




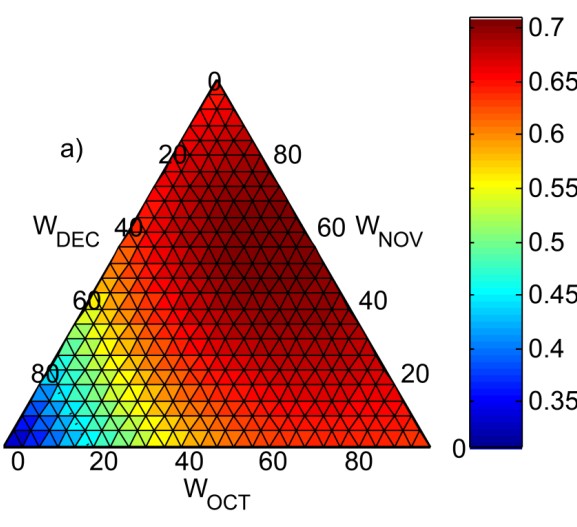

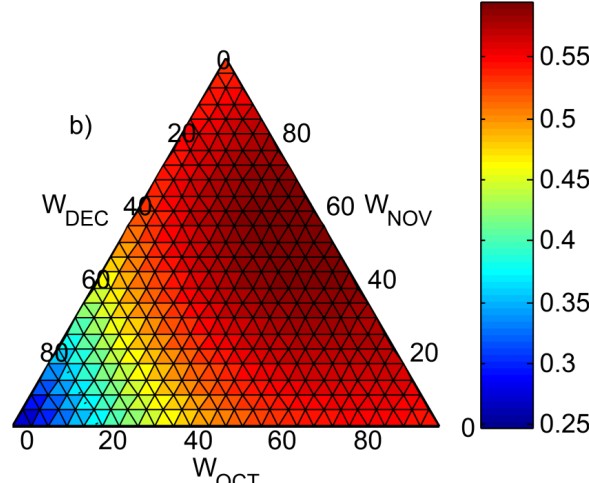

**Figure 5.** The impact of monthly weighting factors in (4) on the sampled correlation between: a) $d\theta_{PM}/dt$ and $P$-$Q$-$E_T$ and b) $d\theta_{PM}/dt$ and $dS_{GR}/dt$.





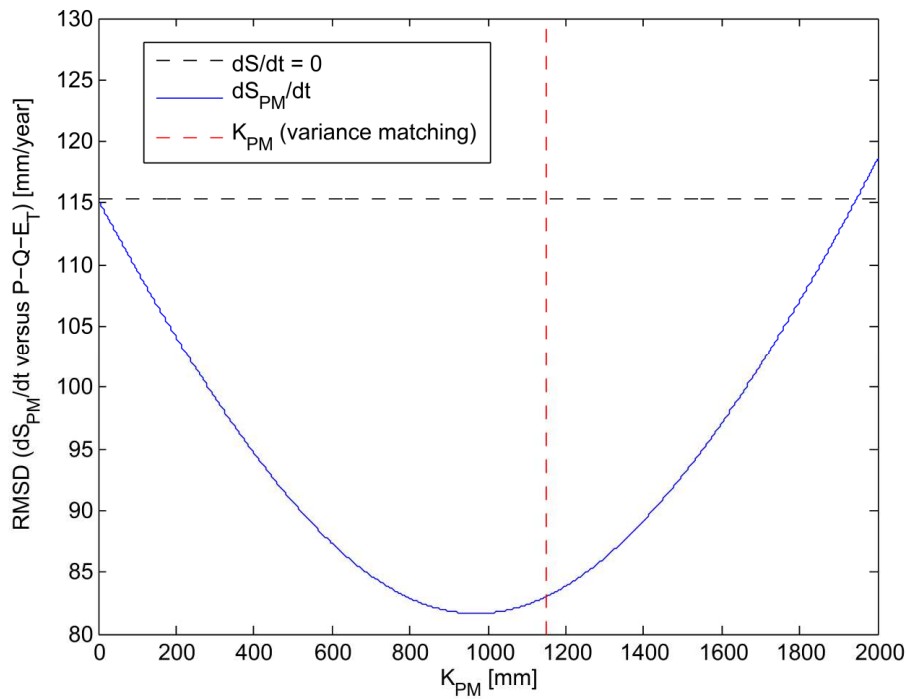

**Figure 6.** The impact of $K_{PM}$ in (5) on the RMSD between $dS_{PM}/dt$ and $P$-$Q$-$E_T$. Also plotted is the standard deviation of $P$-$Q$-$E_T$ (i.e., the RMSD incurred by neglecting annual $dS/dt$) and the value of $K_{PM}$ defined by the

10   variance matching approach in (6).





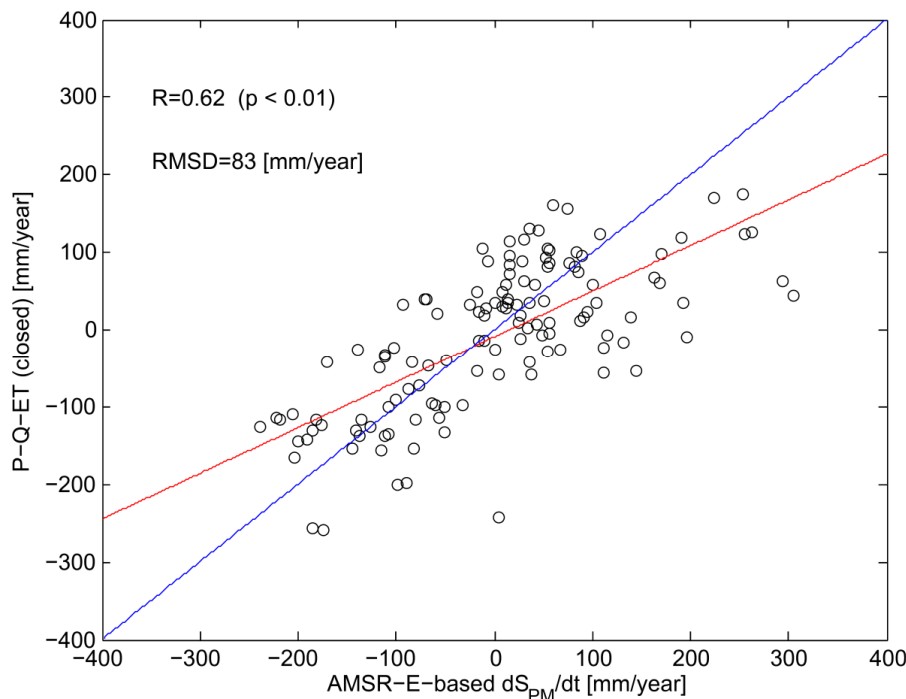

**Figure 7.** Relationship between annual $P\text{-}Q\text{-}E_T$ (closed over the 9-year time series) and the microwave-based $dS_{PM}/dt$ proxy within each of the 16 medium-scale basins listed in Table 1. The blue line is a one-to-one line and

10    red line is the least-squares linear fit.





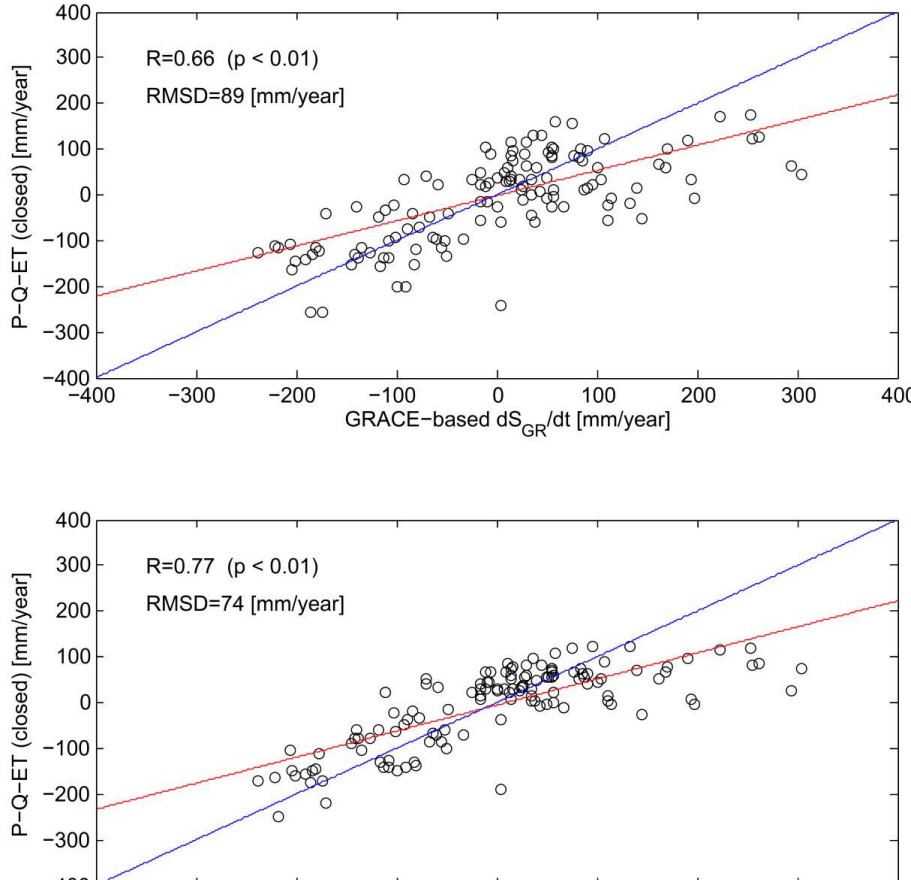

**Figure 8.** a) Relationship between annual *P-Q-E_T* (with 8-year closure) and gravity-based $dS_{GR}/dt$ estimates within each of the 16 medium-scale basins listed in Table 1. Part b) is the same as a) except for correlation against the simple average of $dS_{PM}/dt$ and $dS_{GR}/dt$. The blue line is a one-to-one line and red line is the least-squares linear fit.





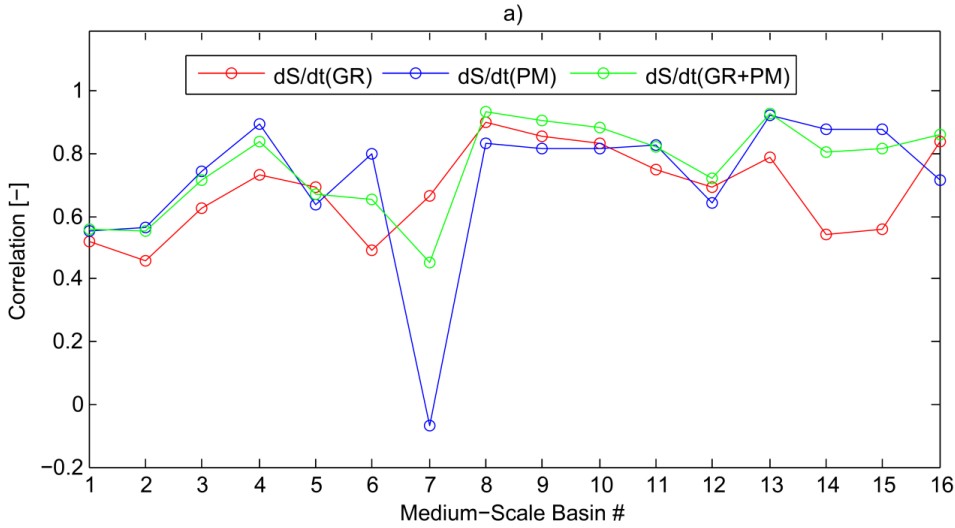

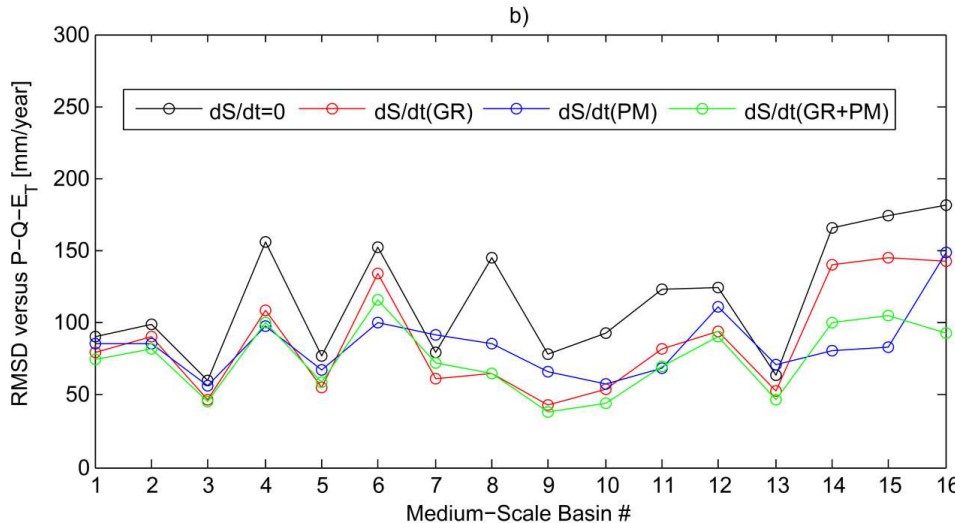

**Figure 9.** For the 16 medium-scale basins listed in Table 1: a) the linear correlation between annual *P-Q-E_T* and various annual *dS/dt* estimates and b) the RMSD between *P-Q-E_T* and various *dS/dt* estimates. Basins are ordered from drier to wetter (from left to right) and basin numbering corresponds to listing in Figure 2.

5    .