# Peer review of "Estimating Annual Water Storage Variations in Medium-Scale (2,000-10,000 km²) Basins Using Microwave-based Soil Moisture Retrievals"

_Hydrology and Earth System Sciences, 2016_

## Referee Comment (RC1) · Anonymous Referee #1 · 15 Dec 2016

This work approaches a very challenging issue: how far we are from a satellite-based estimation of every term of the terrestrial water balance? Probably this is the underlying idea of the authors, but for some reason the focus of the paper, as reflected by the title, is switched to the possibility of estimating water storage from microwave-based surface water content. May be the intention is still there, and this paper represents just a first piece of results. However, it would be useful to clarify the original idea behind the concepts presented here; alternatively, it should be emphasized the relevance of thermal data in context of this study. In a similar way, the calendar year aggregation deemed as questionable by the authors themselves (pag.16, Discussion) appears as an "exit-strategy" following a monthly-scale analysis that provided unsatisfactory results. It would have been reasonable to support the period of temporal aggregation with some considerations about the hydrological yearly cycle in each basin. At the

end, the overall impression is that the authors tried in every possible way to extract a similarity between Grace and AMSR-E datasets, and they finally got it. To this extent, the paper is valuable, and it is able to bring new knowledge, even if the fee paid to the empiricism is probably too high.
* * *

---

## Referee Comment (RC2) · Anonymous Referee #2 · 24 Jan 2017

The paper describes an analysis to use microwave based surficial estimates of soil moisture for annual water storage variations in medium scale catchments. The paper is very well structured, has a strong methodological set-up, written very clearly and scientifically novel. Furthermore, I liked the focus on the applicability of the current RS products for water resources research. To me, the main conclusion is that microwave based soil moisture seems a very good product to downscale GRACE water storage variations measurements to use in 10-fold smaller catchments then currently feasible.

I think the paper is thorough and the authors deserve compliments for their open, critical, step-wise analysis. I think the paper is ready to publish with only a few minor points to address. - Title: I think the title is somewhat too broad. I would prefer the title also states you are looking at medium scale catchments (just add in medium-scale catchments to the title) - Although you make several remarks on the use of a calendar year

instead of a hydrological year and discuss it in section 5, I think especially this parts could benefit from some extra details. Maybe you could add some of the "Preliminary sensitivity analysis . . ." (P17L5) data to the paper. Maybe even a complete sub-section under section 4. Your approach (eq.4-6) is fully empirical, testing a pseudo-hydrological year and e.g. using less months (or add one month with more vegetation but with lower weight factor) seems so logical. - You clearly show the value of microwave soil moisture for downscaling GRACE information. Can you discuss whether there are other downscaling possibilities to have the same effect on GRACE derived storage variations?

Minor edits: P1L18: "contain significant" maybe it is more accurate to say "contain statistically significant" P2L5: certain regions: please specify P2L12 steam > stream P2L19: Q and P are not products but ground based data. Please rephrase P7L18: closure of (1). For me it would be easier if you insert 'equation' between "of (1)". P9L20: snow/could >snow/cold P11L9: suggestion to remove ( ) around "credible"

---

## Author Response (AR1)

**Response to Reviewer #1**

We thank the Reviewer #1 for their useful comments. These comments highlighted some areas of weakness within the discussion paper.

**Reviewer quote #1:**

This work approaches a very challenging issue: how far we are from a satellite-based estimation of every term of the terrestrial water balance? Probably this is the underlying idea of the authors, but for some reason the focus of the paper, as reflected by the title, is switched to the possibility of estimating water storage from microwave-based surface water content. May be the intention is still there, and this paper represents just a first piece of results. However, it would be useful to clarify the original idea behind the concepts presented here

**15 Response:**

The motivation (or in the words of the reviewer "the original idea") for estimating surface water storage from microwave remote sensing is laid out in the first paragraph of the original discussion paper. To quote:

20

5

"Within the past decade, the analysis of data products from the Gravity Recovery and Climate Experiment (GRACE) satellite mission (Tarpley et al., 2004a; 2004b) has led to an enhanced appreciation of the role played by inter-annual variations of total terrestrial water storage (S) within the terrestrial water budget (Chen et al., 2009; Rodell et al., 2007; Syed et al. 2008). However, the

- 25 application of GRACE S retrievals is potentially limited by their extremely coarse spatial resolution (~200,000 km2). In contrast, microwave-based surface soil moisture ( $\theta$ ) retrievals can be obtained at relatively finer resolutions (typically ~1,000 km2). However, such retrievals are hampered by both shallow vertical support (reflecting soil moisture conditions only in the top several centimeters of the soil column) and substantially-reduced accuracy for dense vegetative cover. As a result, they are
- 30 generally assumed to be of limited value for examination of S variations and commonly neglected in water budget studies. However, recent empirical work demonstrates that microwave-based  $\theta$  retrievals are well correlated with GRACE-based S estimates in certain regions (Abelen et al., 2013; 2015). This suggests that  $\theta$  retrievals retain some value for water-balance studies - particularly at spatial scales finer than the resolution of GRACE products."
- 35 Or stated more concisely: 1) gravity remote sensing has revealed that inter-annual variations in terrestrial water storage are important; 2) gravity remote sensing suffers from severe resolution limitations; and 3) microwave remote sensing of soil moisture offers a potential approach for providing higher-resolution assessments. This is the rationale behind looking at microwave remote sensing.
- 40 We feel this rationale is laid out clearly early in the manuscript. However, it could perhaps be (re-)emphasized more throughout the entire manuscript to address the confusion noted by the reviewer.

New text has been added to the first paragraph of Section 4 to ensure that these objectives are alluded to throughout the paper (and not just at the start).

**Reviewer quote #2:**

**...alternatively, it should be emphasized the relevance of thermal data in context of this study.**

5

**Response:**

This point is directly addressed by the second paragraph in the original discussion paper. To quote:

- 10 "Confirming such potential will require the availability of accurate terrestrial water flux variables. Recent progress in the remote sensing of S and  $\theta$  has been mirrored by the increased consideration of satellite-derived evapotranspiration ( $E_T$ ) retrievals in a water balance context (Senay et al., 2011; Hain et al., 2015; Hendrickx et al., 2016; Wang-Erlandsson et al., 2016). In particular, when combined with precipitation (P) and basin-outlet steam flux (Q) measurements, satellite-derived  $E_T$  estimates can be
- 15 used to verify estimates of S variations (dS/dt) obtained from various independent sources (Han et al., 2015). This opens up the possibility for the objective "top-down" evaluation of dS/dt estimates obtained from various remote sensing sources and the opportunity to empirically confront "bottom-up" expectations for these products based solely on theoretical considerations."

Or stated more concisely, thermal-based remote sensing observations are needed to provide

20 evapotranspiration estimates which - when combined with rainfall and stream flow measurements - can be used to independently verify estimates of terrestrial water storage variations obtained from various remote sensing sources.

Again, we feel that these first two paragraphs of the discussion paper directly address the overarching motivation issues raised by the reviewer. New text has been added to the first paragraph of Section 4 to ensure that these objectives are alluded to throughout the paper (and not just at the start).

**Reviewer quote #3:**

In a similar way, the calendar year aggregation deemed as questionable by the authors themselves (pag.16, Discussion) appears as an "exit-strategy" following a monthly-scale analysis that provided unsatisfactory results.

**30 **Response:**

25

The reviewer is misunderstanding our point on page 16 (of the original discussion paper) regarding the use of calendar year averaging. Our point here is not to undercut the motivation for an analysis of interannual water storage variations, rather to acknowledge that there is some sensitivity to the particular set 35 of "book-end" months used to define a year (i.e., January 1 to December 31 versus June 1 to July 30). This was simply done to acknowledge a potential source of sensitivity in inter-annual results and not to underline the value of inter-annual results in general.

In fact, the impact of inter-annual terrestrial water storage variability on the terrestrial water cycle is an area of significant scientific interest. See, for example, recent work aimed on the detection of decadal-5 scale variability in terrestrial storage due to long-term meteorological drought and patterns of anthropogenic ground water extraction or work on the role of groundwater in modulating the impact of climate trends on the hydrologic cycle. These are important scientific issues which can be largely addressed via the measurement of inter-annual water storage variations. This can (and should) be emphasized more in the discussion paper.

10 Obviously, improved temporal resolution (down to e.g. monthly) would be useful in some cases. However, it is unfair to characterize the resolution of inter-annual variations as a fall-back "exitstrategy" meant to mask a failure to achieve a more important goal. The characterization of inter-annual variability is a key goal in and of itself.

The above points are now clarified via new text added to the last paragraph of Section 5.

**15 **Reviewer quote #4:**

It would have been reasonable to support the period of temporal aggregation with some considerations about the hydrological yearly cycle in each basin.

**Response:**

20 This is a fair point. Ideally, the period of temporal aggregation would have been based on hydrological considerations. However, there is an important practical issue to consider. Preliminary analysis suggests that adequately capturing monthly variations requires seasonally and spatially-varying parameters (to capture the relationship between surface soil moisture and terrestrial water storage). Given the (quite-limited) effective sampling size at our disposal (i.e., 8 years), it quickly becomes impossible to adequately calibrate and validate such a high-parameter approach. So while we suspect that a finer (e.g. seasonal) scale approach is possible, we simply lack the data to adequately validate it. This point was already made in Section 5 of the original discussion paper; however, we have revised this section (see

**Reviewer quote #5:**

30 At the end, the overall impression is that the authors tried in every possible way to extract a similarity between Grace and AMSR-E datasets, and they finally got it.

last paragraph of Section 5) to further clarify this key point.

**Response:**

This is not a fair impression (although we acknowledge that weaknesses in our write-up may have contributed to it).

As described in discussion paper, we "tried" only two operations (i.e. linear smoothing and temporally lagging) to resolve both monthly and inter-annual water storage variations (*dS/dt*). Both operations were applied via only two parameter degrees of freedom (i.e. the application of 3 monthly weighting parameters constrained to sum to one).

Based on our attempts, we did not feel like we could adequately validate the monthly approach and stated that conclusion clearly in the original discussion paper (see above and Section 5 of the write-up).

In contrast, inter-annual dS/dt estimates derived from both water balance consideration and GRACE are actually extremely robust. This point was made in Section 4.2 of the original discussion paper:

- 15 "Our primary goal is determining the potential for explaining observed annual P-Q-ET variations in Figure 4 using the microwave-based dSPM/dt proxy introduced above. Our first priority is empirically evaluating the assumptions expressed in (4-6) which underlie the proxy. The first issue is the degree to which the appropriate temporal averaging of microwave-based soil moisture via (4) can be used to obtain a robust linear proxy for P-Q-ET. Figure 5a addresses this by plotting the average linear 20 correlation for all the medium-scale basins between annual P-O-ET and dθPW/dt obtained using all
- 20 correlation for all the medium-scale basins between annual P-Q-ET and dθPM/dt obtained using all potential combinations of WDec. WNov and WOct (where WDec + WNov + WOct = 1.0). Plotted correlations in Figure 5a are generally greater than 0.50 [-]. In fact, even after realistically accounting for the impact of over-sampling due to spatial and temporal auto-correlation in the P-Q-ET fields (Section 2.3), sampled correlations are statistically-significant (one-tailed, 95% confidence) for all possible
   25 combinations of WDec, WNov and WOct."

To summarize, AMSR-E is transformed into a proxy representation of inter-annual dS/dt ( $d\theta_{PM}/dt$ ) via the application of only three monthly weighting parameters (constrained to sum to one). All possible combinations of these parameters lead to an expression of  $d\theta_{PM}/dt$  which has a statistically-significant relationship with (independent) basin-scale measurements of rainfall minus evaptotranspiration minus stream flow.

Therefore, this is not a result that needs to be aggressively "extracted." It is a robust relationship which emerges from any parameterization of a simple weighted average. Also, given the important role of inter-annual water storage variations in a number of research and water resource application issues, it is not a conclusion which can be fairly characterized as a "fall-back" consolation prize. We feel like the

original discussion paper made this point adequately.

**Reviewer quote #6:**

30

To this extent, the paper is valuable, and it is able to bring new knowledge, even if the fee paid to the empiricism is probably too high.

**Response:**

We are unclear what empirical "price" is actually being paid here. As described above the proposed

5 empirical relationship (between surface soil moisture and annual *dS/dt*) is simple, robust, and statistically-significant (when applied appropriately at an inter-annual time) scale. It is a robust empirical "top down" result which will potentially shape our "bottom-up" understanding of large-scale processes linking surface soil moisture with deeper hydrologic units As such it provides a "dividend" rather than paying a "price."

**Response to Reviewer #2**

We thank the reviewer for their generally positive comments and useful suggestions. With only a few minor exceptions (noted below), we agree with all their comments and have modified the manuscript accordingly.

**Review Quote #1:**

Title: I think the title is somewhat too broad. I would prefer the title also states you are looking at medium scale catchments (just add in medium-scale catchments to the title)

**Response:**

Good point. We have modified our title accordingly.

15

5

**Reviewer Quote #2:**

Although you make several remarks on the use of a calendar year instead of a hydrological year and discuss it in section 5, I think especially this parts could benefit from some extra details. Maybe you could add some of the "Preliminary sensitivity analysis data to the paper. Maybe even

20 Maybe you could add some of the "Preliminary sensitivity analysis data to the paper. Maybe a complete sub-section under section 4.

**Response:**

25 We agree - this was a weak link in our original write-up. In response, we have significantly expanded the discussion of this issue in Section 5 of the revised manuscript. In particular, we present new sensitivity results which demonstrate that key manuscript conclusions are not significantly impacted by the particular set calendar months used to define a year.

**30 Reviewer Quote #3:**

Your approach (eq.4-6) is fully empirical, testing a pseudo-hydrological year and e.g. using less months (or add one month with more vegetation but with lower weight factor) seems so logical.

**35 **Response:**

The impact of removing one month is already described in the tertiary plot in Figure 5 (by simply setting the weighting of one particular month to zero). However, the reviewer is correct that there is no analogous discussion concerning the impact of adding one more month. Therefore, in response, new

40 text has been added to Section 4.2 of the revised manuscript which specifically discusses the impact of adding an additional month to equation (4) on optimization results presented in Figure 5.

**Reviewer Quote #4:**

You clearly show the value of microwave soil moisture for downscaling GRACE information. Can you discuss whether there are other down-scaling possibilities to have the same effect on GRACE derived storage variations?

**Response:**

- 10 The revised manuscript now describes alternative strategies for downscaling GRACE-based water storage estimates (see new text added to the first paragraph of Section 1). However, an objective comparison with these earlier strategies is beyond the scope of this paper. The need for such a comparison; however, is now noted in the revised manuscript (see 2nd-to-last paragraph in Section 6 of the revised manuscript).
- 15

**Reviewer Quote #5:**

**P1L18: "contain significant" maybe it is more accurate to say "contain statistically significant"**

20 Response:

Agreed. Changed.

**Reviewer Quote #6:**

25

**P2L5: certain regions: please specify**

**Response:**

- 30 The spatial variation of results in Abelen and Seitz, 2013; Abelen et al. 2015 is difficult to summarize briefly. However, we agree that simply referring to "certain regions" (without providing any specifics) is likely to cause frustration on the part of the readers. Therefore we have modified the sentence to remove any reference to the geographic variation of results these studies (which is not relevant for the current study).
- 35

**Reviewer Quote #7:**

P2L12 steam > stream

**40 **Response:**

Fixed.

**Reviewer Quote #7:**

**P2L19: Q and P are not products but ground based data.**

5 **Response:**

To avoid confusion, "products" has been changed to "datasets."

**Reviewer Quote #8:**

**10**

Please rephrase P7L18: closure of (1). For me it would be easier if you insert 'equation' between "of (1)".

**Response:**

15

Agreed...our original phrasing was awkward. It has been rephrased along the lines suggested by the reviewer.

**Reviewer Quote #9:**

20

P9L20: snow/could >snow/cold P11L9: suggestion to remove () around "credible"

**Response:**

25 Removed.

[revised manuscript text omitted]
Regola lo spazio tra testo asiatico e in
alfabeto latino, Regola lo spazio tra
caratteri asiatici e numeri |
|    | Owe, M., de Jeu, R.A.M., and Holmes, T.: Multisensor historical climatology of satellite-derived global land surface moisture, J. Geophys. Res., 113, F01002, doi:10.1029/2007JF000769, 2008.                                                                                 |              |                                                                                                                                                   |
| 15 | Reager, J.T., Thomas, A.C., Sproles, E.A., Rodell, M., Beaudoing, H.K., Li, B., and Famiglietti, J.S.:                                                                                                                                                                        | <            | Formattato: Tipo di carattere: 11 pt                                                                                                              |
| 15 | Assimilation of GRACE terrestrial water storage observations into a fand surface model for the assessment of regional flood notential. Remote Sens. 7, 14663-14679, 2015                                                                                                      |              | Formattato: Tipo di carattere: 11 pt                                                                                                       |
|    | regionar nood potentiar, remote bens, 7, 14003-1407, 2013.                                                                                                                                                                                                                    |              | Formattato: Tipo di carattere: 11 pt                                                                                                       |
|    | Rodell M., Chen, J., Kato, H., Famiglietti, J.S., Nigro, J.D., and Wilson, C.R.: Estimating groundwater storage                                                                                                                                                               | $\mathbb{N}$ | Formattato: Tipo di carattere: 11 pt                                                                                                              |
|    | changes in the Mississippi River basin (USA) using GRACE, Hydrogeol. J.,-15(1), 159–166, 2007.                                                                                                                                                                                |              | Formattato: Tipo di carattere: 11 pt,
Non Corsivo                                                                                              |
| I  | Rodell, M., Velicogna, I., and Famiglietti, J.S.: Satellite-based estimates of groundwater depletion in India,                                                                                                                                                                |              | Formattato: Tipo di carattere: 11 pt                                                                                                              |
| 20 | Nature, 460, 9991002, doi:10.1038/nature08238, 2009.                                                                                                                                                                                                                          |              | Formattato: Tipo di carattere: 11 pt,
Non Corsivo                                                                                              |
| I  | Saha, S. et al.: The NCEP Climate Forecast System Reanalysis, Bulletin of the American Meteorological Society, 91, 1015, 1057, 2010                                                                                                                                           |              | Formattato: Tipo di carattere: Non
Corsivo                                                                                                     |
|    | y 1,1015 1057,2010.                                                                                                                                                                                                                                                    |              | Formattato: Tipo di carattere: Non                                                                                                                |
|    | Semmens, K.A., Anderson, M.C., Kustas, W.P., Gao, F., Alfieri, J.G., McKee, L., Prueger, J.H., Hain, C.R., Cammalleri, C., Yang, Y., Xiz, T., Sanchez, L., Alsina, M.M., and Velez, M.: Monitoring daily                                                                      |              |                                                                                                                                                   |
| 25 |                                                                                                                                                                                                                                                                               |              |                                                                                                                                                   |
| 25 | evaporatranspiration over two California vineyards using Landsat 8 in a multi-sensor data fusion approach, Rem.                                                                                                                                                               |              |                                                                                                                                                   |
|    | 5015. Environ, in press, 2010.                                                                                                                                                                                                                                                |              |                                                                                                                                                   |
|    | Senay, G.B., Leake, S., Nagler, P.L., Artan, G., Dickinson, J., Cordova, J.T., and Glenn, E.P.: Estimating basin scale evapotranspiration (ET) by water balance and remote sensing methods, Hydrol. Process., 25, 4037–4049. doi:10.1002/hyp.8379, 2011.                      |              |                                                                                                                                                   |
| 30 | Swenson, S., Famiglietti, J., Basara, J., and Wahr, J.: Estimating profile soil moisture and groundwater variations using GRACE and Oklahoma Mesonet soil moisture data, Water Resour. Res., 44, W01413, doi:10.1029/2007WR006057, 2008.                                      |              |                                                                                                                                                   |
|    | Swenson, S.C.: GRACE monthly land water mass grids NETCDF RELEASE 5.0. Ver. 5.0. PO.DAAC, CA, USA. Dataset accessed 2015-01-12 at http://dx.doi.org/10.5067/TELND-NC005, 2012.                                                                                                |              |                                                                                                                                                   |

[revised manuscript text omitted]